# From Gene to Protein—How Bacterial Virulence Factors Manipulate Host Gene Expression During Infection

**DOI:** 10.3390/ijms21103730

**Published:** 2020-05-25

**Authors:** Lea Denzer, Horst Schroten, Christian Schwerk

**Affiliations:** Department of Pediatrics, Pediatric Infectious Diseases, Medical Faculty Mannheim, Heidelberg University, 68167 Mannheim, Germany; lea.denzer@medma.uni-heidelberg.de (L.D.); horst.schroten@umm.de (H.S.)

**Keywords:** virulence factors, bacteria, host-pathogen interaction, gene expression, immune response, manipulation, inflammation, persistence, replicative niche

## Abstract

Bacteria evolved many strategies to survive and persist within host cells. Secretion of bacterial effectors enables bacteria not only to enter the host cell but also to manipulate host gene expression to circumvent clearance by the host immune response. Some effectors were also shown to evade the nucleus to manipulate epigenetic processes as well as transcription and mRNA procession and are therefore classified as nucleomodulins. Others were shown to interfere downstream with gene expression at the level of mRNA stability, favoring either mRNA stabilization or mRNA degradation, translation or protein stability, including mechanisms of protein activation and degradation. Finally, manipulation of innate immune signaling and nutrient supply creates a replicative niche that enables bacterial intracellular persistence and survival. In this review, we want to highlight the divergent strategies applied by intracellular bacteria to evade host immune responses through subversion of host gene expression via bacterial effectors. Since these virulence proteins mimic host cell enzymes or own novel enzymatic functions, characterizing their properties could help to understand the complex interactions between host and pathogen during infections. Additionally, these insights could propose potential targets for medical therapy.

## 1. Introduction

Successful defence against extracellular and intracellular bacteria primarily relies on the ability of innate immune cells to sense present bacteria followed by activation of the adequate matching immune response. The identification of bacteria is enabled by a broad array of pathogen recognition receptors (PRRs), which recognize extensively conserved pathogen-associated molecular patterns (PAMPs) as nucleid acids, cell wall components and proteins from viruses, bacteria, fungi and parasites [1]. Toll like receptors (TLRs) and NOD-like receptors (NLRs) represent the two major classes of PRRs, acting at the cell surface or in the cytoplasm, respectively [2]. After activation, PRRs induce multiple signalling pathways aiming at the expression of proinflammatory cytokines, which regulate the innate immune response. The most prominent pathways involved are mediated by mitogen-activated protein kinases (MAPKs) or nuclear factor-κB (NF-κB), which are initiated by path-specific adapter proteins and transferred by downstream phosphorylation cascades [3]. Interestingly, all pathways can synergize to guarantee a specific and appropriate immune response for the present pathogen. This is enabled by the high diversity of PRRs, PAMPs and PRR adapter proteins and their numerous ways to be combined during innate immune response [1,4].

The immune response has to be tightly controlled to ensure a clearance of the bacteria but also to prevent tissue damage and necrosis as result of sepsis. There are several levels to influence the expression of inflammatory genes. A first level of interference is changing of the DNA’s structure on the chromatin level. Epigenetic modulation enables remodelling of the chromatin to transfer heterochromatin into euchromatin allowing transcription or vice versa [5,6,7]. In addition, the affinity of promotors and other regulatory DNA sequences for RNA polymerases and transcription factors (TFs) can be influenced by cytosine or adenine methylation. To induce transcription, TFs and RNA polymerases are recruited to target genes, a step that represents another level to regulate gene expression. Only a minor portion (fewer than 2%) of genes is transcribed into mRNAs, instead the majority is transferred into so called non-coding RNAs (ncRNAs). The long ncRNAs (lncRNAs) as well as some classes of short ncRNAs are also involved in epigenetic regulations but its most studied group, the miRNAs (microRNA), are mostly involved in RNA destruction [8].

Another point for interference with gene expression is during processing of mRNAs, which includes 5′ capping, alternative splicing and polyadenylation [9,10]. Primary ncRNAs can be processed by the protein complex DICER (eukaryotic ribonucleases) to generate miRNAs, which can negatively regulate expression of its primary transcript. To guarantee proper cell function, mRNAs need to be degraded after a certain time frame; the RNA stability is, therefore, another switch to modify gene expression. Polyadenylation and 5′capping prolongs RNA stability but several enzymes are able to decap the 5′cap and to remove the polyadenyl tail of the RNA leaving an unprotected mRNA [11,12,13]. Nevertheless, these structures are crucial for translation initiation and can gain enough time for the mRNA to be translated. This represents a further step for interference with gene expression, as there is the need of several factors to induce and prolong translation [14]. Translation initialisation factors have to be recruited leading to ribosome assembly and binding of the first amino acid loaded tRNA (transfer RNA). To keep translation ongoing, elongation factors and ATP (Adenosine triphosphate) have to be present. The nascent protein chain then needs to fold into its physiological form to be active [15,16]. Therefore, protein-folding and activation catalysed by different chaperones is another important step during gene expression and is also tightly controlled. Finally, a last step to regulate gene expression is represented by the stability and degradation of, in some cases misfolded, proteins [17,18]. An overview of the steps during host gene expression targeted by bacterial pathogens, as well as the bacteria involved, is given in Figure 1.

Host cells fight bacteria with a proinflammatory cytokine response, lysosomal degradation, autophatic clearance and activation of the unfolded protein response, which in the end can lead to apoptosis [19]. Bacteria can hijack all these defence mechanisms by interfering with the host’s gene expression at any level. For that purpose, bacteria express several virulence factors, the so-called effectors. These proteins are able to mimic host enzymes, thereby manipulating the host response following invasion favouring intracellular survival, persistence and spreading. Since proteins and enzymes of signal transduction pathways are involved in all defence mechanisms, they are a favoured target of effector proteins [19]. It is worth noting that other virulence proteins own novel enzymatic functions, which allow them to enter the nucleus and directly induce gene expression or repression. Therefore, these bacterial effectors are termed nucleomodulins [20,21]. In the following we will present an overview on the manipulation of host gene expression at different levels by nucleomodulins and other bacterial effectors.

## 2. Bacterial Virulence Factors Manipulating Host Gene Expression

### 2.1. Epigenetic Control of Gene Expression

The expression of genes is dependent on their accessibility for RNA polymerase II (RNA Pol II) and TFs. As approximately 147 base pairs of the DNA are wrapped around histone octamers build by the subunits H2A, H2B, H3 and H4 as well as the scaffold protein H1 to form the nucleosome, those sequences are protected from transcription [5,6,7]. Therefore, the packaging of the nucleosomes defines the chromatin state into euchromatin and heterochromatin enabling transcription or blocking it. In order to react properly to a certain stimulus, the chromatin state can be remodeled to give access to the required genes, a process called nuclear remodeling or histone modification. Enzymes posttranslationally modify the amino acids at the N-termini of the histone proteins (called histone tails) by acetylation, phosphorylation, methylation and ubiquitination in a reversible manner to modify the interaction between neighbored nucleosomes favoring an open or closed chromatin state [19,20,22]. Nucleosomes are then allowed to slide along the chromatin fiber in an ATP-dependent manner, to give access to the DNA sequence. This reveals the dual function of chromatin, to provide a natural scaffold and being part of an essential regulatory signaling network processing the incoming data to create a special transient biological output [23]. On top, established posttranslational modifications (PTMs) can be maintained beyond the initial signal and cell divisions inheriting cell type specific gene expression enabling cell lineage specification and cellular identity [23,24,25].

The enzymes responsible for the modulation of the histone tails are divided into “writers,” which attach the chemical units, “readers,” which recognize and translate them by recruitment of activating or repressing factors and “erasers,” which remove the modifications. The resulting “epigenetic code” is highly dynamic, as each established modification influences the addition or removement of other modifications that in turn influence the own stability and persistence. Moreover, epigenetic mechanisms represent the missing link between more or less stable gene expression and the impact of environmental factors on gene expression that can also cause diseases as cancer [5,26,27]. Therefore, these enzymes represent a central role in the regulation of immune responses as alterations in their activity and expression profiles leading to global changes in the histone modification pattern have been detected as cause of several chronic immune diseases as asthma, chronic obstructive pulmonary disease, colitis, systemic lupus erythematosus and rheumatoid arthritis [28,29].

Additionally, the DNA can be methylated at cytosine or adenosine residues converting them into methyl-cytosine or methyl-adenosine to cause transcriptional repression [30]. Hypermethylation dominantly occurs at CpG islands, cytosine-guanine rich regions at promotor regions, disrupting TFs and RNA polymerase binding to DNA or recruiting other co-repressors. A hypermethylated gene, that was not methylated before is therefore, not suitable for transcription and with the recruitment of further silencing-factors, will finally be silenced. This kind of modification is thought to provide a stable gene silencing that can be inherited to the next generation of cells [31,32].

Moving away from the old definition of epigenetics as hereditable stable changes at chromatin and DNA without changing its sequence, modern opinion changed towards a highly dynamic and reversible mechanism of gene regulation also enabling short term adaptions to changing environments [30]. As consequence, regulation through ncRNAs are also included to the epigenetic regulatory repertoire that can be classified according to their length into short ncRNAs (<200 nucleotides), which include miRNAs or long ncRNAs (>200 nucleotides) [33].

#### 2.1.1. Manipulation at the Level of Histone Modifications

After recognition of bacterial presence by PRRs, signaling cascades activate proinflammatory cytokine expression. To improve accessibility of TFs, such as NF-κB, to the promoters of inflammatory response genes, an activating histone modification as phosphorylation of Serine 10 on histone H3 (H3S10) is established, which itself is mediated by MAPK signalling. It has been shown, that the virulence factor LPS alone is able to induce a global increase of H3S10 leading to promotion of gene expression proving the high sensitivity of the immune reaction [34].

Recent studies revealed that bacteria directly interfere with a host’s histone modifications to dampen the expression of proinflammatory cytokines by the secretion of effectors. Presence of *Listeria monocytogenes* induces phosphorylation of H3S10 but the bacterium is able to remove this activating phosphorylation within short time [35,36,37,38]. The secreted virulence factor Listeriolysin (LLO) mediates this mechanism and is also responsible for a global deacetylation of H3 and H4. Other bacteria, as *Clostridium perfringens* or *Streptococcus pneumoniae*, produce toxins, such as perfringolysin and pneumolysin, respectively, that belong to the same family as LLO and show also a similar effect on H3S10 phosphorylation [36]. The decreased levels of phosphorylated H3S10 and acetylated H4 at proinflammatory genes resulted in transcriptional downregulation thereby damping the immune response. As this observation is only dependent on the membrane-binding ability of LLO, it is most likely that LLO modulates the signal transduction to induce alterations in the histone modification pattern [36].

Like *L. monocytogenes*, *Shigella flexneri* is also able to inhibit H3S10 phosphorylation by secretion of phosphothreonine lyase effector OspF, which dephosphorylates MAPKs as p38 or ERK resulting in attenuated NF-κB binding at promotors of inflammatory genes [39]. Together with OspB, another effector of *Shigella,* OspF, interacts with the human retinoblastoma protein Rb that is capable of binding several chromatin-remodeling factors [40,41]. In this constellation, *Shigella* adjusts the chromatin structure at specific genes to downregulate host innate immunity.

*L. monocytogenes* owns another effector, which induces deacetylation on lysine 18 of histone H3 (H3K18). Thereby, Internalin B (InlB) activates the host histone deacetylase sirtuin 2 (SIRT 2), leading to repression of transcriptional start sites through occupation by SIRT 2 and following downregulation of the immune response, which could be attenuated by SIRT 2 inhibition [42]. The listerial virulence factor LntA enters the nucleus after infection of epithelial cells targeting the chromatin silencing complex component BAHD1. Together with heterochromatin protein 1 (HP1), methylated DNA-binding protein 1 (MBD1), histone deacetylases (HDAC1/2) and the KRAB-associated protein 1 (KAP1/TRIM28) that are involved in heterochromatin formation, BADH1 targets interferon-stimulated genes (ISG) for silencing by binding to their promotors [43,44]. This is inhibited by LntA, which is thought to promote chromatin-unwinding and as consequence upregulation of ISG by histone H3 acetylation. The exact mechanisms, how BAHD1 is recruited to its targets and how LntA interferes with this process has still to be investigated [21].

Another prominent histone modification is the methylation or demethylation of lysine residues, mediated by histone N-lysine methyltransferase (HKMT) or histone demethylases (HDM), respectively. Several bacteria express HKMT effectors, which enable them to directly interfere with host gene regulation as they are mimics of host chromatin modifiers. As there are many HKMT homologues in the repertoire of bacterial effectors described this mechanism seems to be a successful strategy to subvert host gene expression [45]. The nuclear effector (NUE), is secreted by *Chlamydia trachomatis* via a type III secretion system (T3SS) to enable its localization to the nucleus, where it might methylate H2B, H3 and H4. The homologous effectors RomA and LegAS4 secreted by *Legionella pneumophila Paris* and *L. pneumophila Philadelphia Lp02* strains, respectively, methylate H3 to alter host transcription but target different residues [45,46]. RomA represses global transcription by methylation of histone 3 lysine 14 (H3K14), a modification that is known to compete with the activating acetylation of H3K14 [46]. Contrary to RomA, LegAS4 increases transcription of ribosomal RNA genes (rRNA) through methylation of histone 3 lysine 4 (H3K4) but if this modification is mediated by LegAS4 alone it is not clear yet [45]. Interestingly all described bacterial methyltransferases own a conserved SET (Suppressor of variegation, Enhancer of zeste and Trithorax) domain, which uses a S-adenosyl-l-methionine (SAM) methyl donor to catalyze methyl group attachment to lysine residues [45,47]. One example is the effector BtSET, secreted by *Burkholderia thailandensis* that localizes to the nucleolus to methylate histone H3K4 promoting transcription of rRNA genes. Some effectors are capable of more unusual modifications, for example, the effector BaSET identified in *Bacillus anthracis* trimethylates histone H1 but none of the core histones. This effector represses the expression of NF-κB target genes after transient overexpression in mammalian cells and its deletion results in the loss of virulence [45,46,47].

Another modification, which differs from the known mechanisms of histone modification, is represented by dimethylation of histone 3 on arginine 42 (H3R42me2), a residue critical for DNA entry/exit from the nucleosome and not located at the histone N-termini. This modification is involved in the regulation of ROS (reactive oxygen species) production, which represents a crucial host defense mechanism against bacterial pathogens [48]. *Mycobacterium tuberculosis* represses genes involved in ROS production by secreting Rv1988, a methytransferase able to establish H3R42me2 to increase survival in host macrophages [48]. An overview of bacteria and their effectors that are secreted to induce histone modifications is given in Table 1.

Influencing the expression of histone modifying enzymes is another possibility to affect histone modifications in favor of bacterial survival (see Table 2). Modulation of the histone deacetylase HDAC1 appears to be most targeted by pathogens, to manipulate the key acetylation system enabling protection against eradication. Infection with *Anaplasma phagocytophilum*, an intracellular pathogen causing human granulocytic anaplasmosis, causes upregulation of HDAC1 leading to a globally increased HDAC activity [56]. The recruitment of HDAC1 to AT-rich chromatin sites in promotors of host defense genes is mediated by the effector ankyrin A (AnkA) resulting in the reduction of histone H3 acetylation and the suppression of target genes such as *CYBB* that encodes Cytochrome b-245, beta polypeptide. As this element of the phagocyte NADPH oxidase is involved in the clearance of the pathogen by neutrophils, it is preferentially targeted [57,58,59,60]. Furthermore, AnkA functionally mimics SATB1, a protein able to bind AT-rich sequences distributed across distinct chromosomes at attachment regions of the nuclear matrix. Proteins with this ability are involved in nuclear matrix attachment, spatial organization of chromatin and large-scale transcriptional regulation [59,61,62,63]. AnkA could also perform as global organizer of the neutrophil genome, thereby acting locally (*cis*) and at a distance (*trans*) to a target gene. Moreover, pathogens as *Chlamydia psittaci* secrete nucleomodulins (SinC) that could act like AnkA and influence anchoring factors and lamins that control the dynamics of chromatin looping and organization, as the inner nuclear membrane proteins MAN1 and LAP1 [64].

*Pseudomonas aeruginosa*, an opportunistic pathogen that infects and colonizes inflamed airways and burn wounds, induces HDAC1 expression in human THP-1 monocytes with the help of a molecule usually used for quorum sensing, 2-aminoacetophenone [65,66]. This is followed by global histone H3K18 hypoacetylation and reduced expression of inflammatory cytokines and chemokines (e.g., TNF, IL-1b and MCP-1) resulting in dampened host defense against the bacterium.

Considering that this effect was also dampened by knockdown or inhibition of class I HDACs and the evidence that besides *A. phagocytophilum* and *P. aeruginosa* also *Porphyromonas gingivalis* modulates HDAC1 during infections, HDAC1 family members might play a central role in development of an epigenetic mediated tolerance against the pathogens [67]. In patients with chronic periodontitis, mRNA and protein levels of HDAC1 expression were globally increased compared to healthy individuals and colocalized with TNF expressing cells and tissues. Interestingly, epigenetic regulation mediated by *P. gingivalis* seems to be cell-type specific, since HDAC1 and HDAC2 are downregulated in gingival epithelial cells *in vitro*, while levels of acetylated histone H3 were increased in murine epithelial cells of the gingival tissue [68,69]. In addition, the host acetylation system is also often influenced by short chain fatty acids (SCAFs) produced by commensals or pathogenic bacteria as *P. gingivalis* (for recent reviews please refer to References [70] and [71]).

Another strategy followed by bacteria during host infection and manipulation of the epigenetic regulatory mechanisms is to proteolytically degrade histone acetyl transferase (HAT) family members. One example of bacteria using this strategy are enteropathogenic and enterohaemorrhagic *Escherichia coli*, which secrete the effector protein NleC, a zinc-dependent metalloproteinase targeting intracellular signaling to dampen the host inflammatory response [76]. The protein specifically binds and degrades the host HAT p300 in infected cells leading to decreased IL-8 production, an effect that can be restored by p300 overexpression. Thus, HATs and HDACs can both be targeted by pathogenic bacteria to modulate epigenetics and inflammatory gene expression in their benefit.

#### 2.1.2. How to Control Host DNA Methylation

DNA methylation is another way to control gene expression. There are several enzymes called DNA-(cytosine C5)-methyltransferases (DNMTs), which establish methyl residues to cytosine or adenosine residues, respectively [71]. In contrast, the removement of DNA methylation patterns is more complex, as the modified nucleotides or DNA sequences have to be exchanged by DNA-repair mechanisms or the methylation has to be oxidized to form 5-Hydroxymethylcytosine, which can be removed by enzymes [77]. DNA methylation patterns at promotors of tumor suppressor genes had already been discovered, when first hints pointed towards an influence of bacterial inflammation on mechanisms establishing DNA-methylation patterns after *Helicobacter pylori* infection. In this context, among others, genes associated with cell growth (*apc, p14 (ARF), p16 (INK4a)*), cell adherence (*cdh1, flnc, hand1, lox, hrasis, thbd, p14ARC*) and DNA-repair (*brca1, mgmt., hMLH1*) are influenced [52,78,79,80]. Similar observations of altered DNA-methylation patterns during inflammation were made following uropathogenic *E. coli, Campylobacter rectus* and *Mycobacterium leprae* infections [81,82,83]. Still, the questions if DNA-methylation is directly induced by bacteria or is a secondary reaction by the host due to persistent inflammations, as well as the underlying mechanisms, are not completely answered yet [84].

However, several *Mycoplasma* species to encode mammalian DNMTs like equivalents that target cytosine-phosphate-guanine (CpG) dinucleotides to establish methylation patterns in the bacterial genome [85,86,87]. Moreover, their expression in human cells results in their translocation to the nucleus, where they set up unusual methylation patterns on the host DNA. This was shown for the DNMTs Mhy1, Mhy2 and Mhy3 expressed by *Mycoplasma hyorhinis* in combination with up- and downregulation of certain genes resulting in activation of proliferation specific pathways, a process that might contribute to tumor progression [85,88].

*Mycobacterium tuberculosis* owns an effector called Rv2699 that can enter the nucleus of THP1 cells (a monocytic cell line derived from a patient with acute monocytic leukemia) and methylate cytosines outside CpG dinucleotides. Notably, Rv2699 prefers cytosine-phosphate-adenine or cytosine-phosphate-thymine sites to generate a type of methylation that is, with few exceptions, normally not present in mammalian adult differentiated cells [89,90]. However, non-CpG methylation could lead to a more stable type of modification that persists longer in the genome of infected nondividing macrophages, offering an advantage for *M tuberculosis* by establishing an intracellular environment for persistence [90]. A follow up study revealed that THP1 macrophages infected with *M. tuberculosis* strain H37Rv created genome-wide *de novo* methylation patterns at non-CpG dinucleotides that included hyper- and hypomethylated regions [90,91]. Additionally, clinical isolates infecting THP1 cells may downregulate IL-6 receptor expression by hypermethylation of CpG-dinucleotides at the promoter of the IL-6 receptor gene. Still, it has to be mentioned, that the observations of *M. tuberculosis* induced DNA-methylation patterns depend on the infected cell type.

Another interesting bacterial induced modification of gene expression is represented by differentiated Swann cells that adapt the phenotype of progenitor stem-like cells after *M. leprae* infection. This is probably induced by silencing of the *Sox10* gene after bacterial methylation [82]. In contrast to the decreased expression of Sox10, other genes involved in epithelial–mesenchymal transition (EMT) were demethylated and transcribed leading to the transformation of Swann cells into myofibers or smooth muscles in vitro and in vivo [92].

*P. gingivalis* was shown to increase the methylation of the TLR-2 promotor in gingival epithelial cells (GECs) reducing innate immunity activation and causing hyposensibility [69,93]. Besides, coinfection with *Filifactor alocis*, another pathogen associated with periodontitis is suggested to influence the whole cell transcriptome through impact on the nucleosome structure by reduced expression of H1 family members [73,74]. Other histone modifications induced by LPS or short chain fatty acids (SCFAs) produced by *P. gingivalis* are summarized in Table 1 and Table 2.

Still, there is not much known about the relation of DNA-methylation and infection and the underlying causalities [71,84]. Considering that many of these modifications are observed in the context of cancer initiation and progression, further investigation may contribute to new therapeutic agents and cancer prophylaxis.

#### 2.1.3. Regulation of Host Gene Expression via lncRNAs

The role of lncRNAs during modulation of gene expression has been discovered in the recent years. Similar to mRNAs, lncRNAs are transcribed by RNA polymerase II or III, followed by splicing, 5′capping and in some cases polyadenylation at the 3′end. Contrary to mRNAs, the expression of lncRNAs is much lower and in a cell-, tissue- and developmental stage-specific manner [94].

Dependent on of their position relative to the neighboring protein-coding gene, lncRNAs are classified as sense, antisense, bidirectional, intronic or intergenic and, despite their enormous number, they were previously considered as “dark matter” or “junk” in the genome [95]. *Au contraire*, lncRNAs are now respected as important physiological regulators during cell homeostasis, growth, differentiation and anti-viral responses [96,97,98,99]. In addition, gene imprinting, regulation of the p53 pathway, stem cell self-renewal and differentiation and DNA damage response were reported as lncRNA controlled mechanisms [100,101,102,103].

The functionality of lncRNAs is not restricted to the neighbored protein-coding gene (*in cis*), in contrast they are also able to act *in trans* to regulate gene expression across chromosomes. In this context, lncRNAs regulate different processes as chromatin remodeling, transcription and post-transcriptional regulation via their capacity as signals, decoys, guides and scaffolds [104,105]. Interestingly, another origin of lncRNAs is the expression of pseudogenes and gaining Influence over the expression of pseudogenes could, therefore, provide a possibility to control infectious responses [106].

Immune regulation through lncRNAs has already been known after viral infections but recent research indicates its involvement also whilst fighting bacteria [107]. In that context, 76 enhancer RNAs (eRNAs), 40 canonical lncRNAs, 65 antisense lncRNAs and 35 regions of bidirectional transcription are differentially expressed in human monocytes after LPS stimulation [108]. LPS stimulation alone induces a differential expression of about 27 lncRNAs leading to histone trimethylation or acetylation of neighboring genes after de-regulation, pointing towards their regulatory influence during the innate-immune response [109]. The observation, that 44% of total lncRNAs varied in their expression after *Salmonella* infection in HeLa cells could foster these results and substantiate them by a function in the early phase of infection as sensitive markers for pathogen activity [110]. In line with this, the lncRNA HOTAIR that contributes to transcriptional repression of HOX genes also promotes inflammation in mice cardiomyocytes by TNF-α production mediated through phosphorylation of p65 protein and NF-κB activation after LPS induced sepsis [111,112].

Long intergenic non-coding RNAs (lincRNAs) are a subtype of lncRNAs, as they are expressed from intergenic regions. In response to an LPS stimulus, bone-marrow dendritic cells expressed about 20 lincRNAs with the majority being dependent on NF-κB activity, including lincRNA-Cox2, which is also upregulated in bone marrow-derived macrophages following *L. monocytognes* infection [113,114]. Additionally, bacteria sabotage lncRNA activity, as BCG (attenuated strain *M. bovis* bacillus Calmette-Guérin BCG) infected macrophages repress the expression of 11 lncRNAs that are not dampened by infection with heat activated bacteria [115]. Still, possible subversion of lncRNA-mediated inflammatory regulation needs to be further investigated.

### 2.2. Bacterial Effectors Manipulating the Host Transcription Machinery

Proper RNA Pol II complex formation is essential for protein expression and tightly controlled by regulators, who are expressed by approximately 10% of all genes [116]. These are general or specific TFs, which serve as activators and repressors and determine specificity and efficiency of transcription at individual promotors [117,118]. Considering the large number of factors involved in transcriptional regulation, it is not surprising that bacteria target those regulators to drive transcription in their favor. For example, activator protein-1 (AP-1)–dependent gene transcription is inhibited by NleD, an effector of *E. coli* that cleaves and inactivates the MAPKs, JNK and p38. Also, the recently identified nucleomodulin OrfX secreted by *L. monocytogenes* that influences host transcription via its interaction with the Ring1 YY1-binding protein (RYBP), a multifunctional nuclear protein owning a zinc finger motif to interact with several TF components of the polycomb repressive complex 1 [119,120]. Moreover, RYBP promotes gene silencing and transcriptional repression of developmental genes, as it is part of the BCL6 corepressor (BCOR) complex [121,122]. In contrast, it is also involved in the activation of the Cdc6 promoter and mediates interaction of the TFs E2F and YY1 [123]. Furthermore, the TF p53 (a tumor suppression factor) is assumed to be stabilized by RYBP through binding of MDM2, an E3 ligase, preventing p53 from proteosomal degradation. As p53 controls intracellular levels of reactive oxygen (ROS) and nitrogen species (RNS) that are part of the immune defense of macrophages, OrfX targets RYBP for degradation to interrupt P53 activity promoting intracellular bacterial survival. Still, this model needs to be verified [120,124].

*Salmonella* Typhimurium inhibits the expression of NF-κB mediated genes by secretion of PipA, GogA and GtgA via its type II secretion system [125,126,127,128]. These proteins belong to the family of zinc metalloproteases and contain the short metal binding-motif HE*XX*H, which consists of two histidine residues coordinating the active-site zinc and a glutamate residue that is essential for catalytic activity [129]. They can cleave NF-κB TF subunits, including p65, RelB and cRel, thereby suppressing their ability to control the transcription of innate immune genes [125,126,127,128]. In addition, Jennings et al. predicted suppression of the transcriptional coactivator ribosomal protein S3 (RPS3) by GtgA family members, as it produces p65 (1–40) after cleavage of p65.

Enteropathogenic and enterohemorrhagic *E. coli* possess another zinc metalloprotease that also owns the HE*XX*H motif and is able to cleave p65, RelB and cRel, as well as NF-κB1 (p105/p50) and NF-κB2 (p100/p52) [126,127,128,130]. Following cleavage, the subunits are left inactive except for the N-terminus of p65, which prevents the nuclear import of the transcriptional coactivator ribosomal protein S3 (RPS3). This in turn inhibits the expression of a specific subset of NF-κB–dependent genes requiring RPS3 for their expression.

The non-pathogenic *E. coli* strain 83972, who is the agent causing persistent asymptomatic bacteriuria (ABU), suppresses host defense in the urinary tract by inhibition of RNA Pol II dependent transcription. While infections with pathogenic strains induce urinary tract infections, these bacteria create an asymptomatic carrier state that reminds of bacterial commensalism and protects patients against infection with more virulent strains. Therefore, therapeutic urinary tract inoculation with the ABU strain is a promising alternative to appease symptoms of therapy resistant, recurrent urinary tract infections [131,132,133].

Studies with patients and human cells treated with ABU strain 83972 revealed that 24 h after inoculation, over 60% of all genes were suppressed, including regulatory elements as transcriptional repressors, transcriptional activators, regulators of translation and chromatin or DNA organizing factors [134,135]. This phenomenon was observed for many genes of the innate immune response but about 22.5% of the effected genes are involved in Pol II transcription or in regulating Pol II–dependent pathways. After Ingenuity Pathway Analysis, a network incorporating *FOSB*, *HSPA6*, *RN7SK*, *RGS4* and *IFIT1,* inversely regulated genes that control Pol II for instance through TATA box–binding proteins (TBP) [136,137,138], appeared.

Another fascinating observation revealed that 50% of ABU strains lack virulence genes due to point mutations or deletions resulting in smaller genome sizes. Considering that ABU strains evolved from uropathogenic *E. coli*, this could be a hint for a reductive evolution creating a niche through active adaptation to the host environment [139]. Thereby, the ABU strain generates a commensal like state characterized by a well-balanced immune environment that finally protects the host from colonialization with more virulent strains and destructive immune activation [134,140,141,142,143,144].

Ambite et al. observed in a follow up study that obtaining one single virulence factor was enough to induce virulence of a non-virulent strain causing symptoms in the host, in contrast to the broad repertoire of virulence factors that are normally expressed by pathogens [145]. In this context, reconstitution of the *papG* adhesin gene recreated functional P-fimbriae leading to virulence of the avirulent ABU strain. Considering the high frequency of ABU strains carrying inactive papG genes, the loss of P-fimbriae might induce development of virulence attenuation and evolution towards commensalism [96,134].

*L. pneumophila* is another pathogen that induces global reprogramming of transcription, by interference with transcriptional elongation by Pol II. Its effector AnkH interacts with LARP7, a component of the 7SK small nuclear ribonucleoprotein (snRNP) complex involved in Pol II pausing. Thereby, the β-hairpin loop of the third ankyrin repeat of AnkH impairs LARP7 interaction with the other 7SK snRNP complex components resulting in promotion of gene wide transcriptional elongation [146]. The nucleomodulin SnlP expressed by *Legionella* also regulates RNA Pol II mediated transcription elongation by inhibition of SUPT5H that is part of the 5,6-Dichlorobenzimidazole 1-β-D-ribofuranoside (DRB, a selective inhibitor of transcriptional elongation by RNA pol II) sensitivity-inducing factor (DSIF) complex [147].

### 2.3. RNA Processing as Target during Infections

Following transcription the immature pre-mRNA is processed to mature mRNA, a process that includes 5′ capping, 3′polyadenylation and splicing and is essential for normal cell function [148]. Splicing of mRNAs can include omitting or retaining of exons to create a protein with altered structure, function and stability, a process called alternative splicing. In the human genome, more than 90% of all genes are adjusted by alternative splicing, which enables a variation and dynamism in the static genome as protein domains can be easily new combined to form isoforms with unique functional abilities [148,149]. The process is controlled by multi-molecular complexes that assemble at splice junctions, thereby evaluating splicing enhancer/silencer elements flanking splice junctions, which in their combination determine inclusion or exclusion of exons. Those elements are divided into *cis* elements including splicing enhancers and silencers and *trans* elements as snRNPs, hnRNPs, SR proteins (serine-arginine containing proteins) and several other accessory proteins [149,150]. Furthermore, the rate of transcription has a critical influence on alternative splicing, as a paused or decelerated RNA Pol II can use newly transcribed splice junctions that could have been skipped at higher translation speed [151,152,153].

Recent studies indicate that the host splicing machinery is targeted by pathogens to perturbate immune response. This has been extensively reported for viral infections but quite less is known about bacterial interference. Nevertheless, global alterations of splicing patterns were detected after infection of human macrophages with *M. tuberculosis, Salmonella* or *Listeria* [154,155,156,157]. More specifically, hnRNP M interacts with LLO leading to a hampered INF-γ response [158] and co-immunoprecipitation of splicing factors hnRNP U, hnRNP H, hnRNP A2/B1 isoform A2 and SRSF3 with the bacterial protein mtrA was shown in macrophages overexpressing specific secreted proteins that are infected with *M. tuberculosis* [159]. Another hint for the interaction of mycobacterial proteins and host splicing factors is the precipitation of host splicing proteins as SRSF2, SRRM2, SF1, HTATSF1, GCN1L1, CPSF6 and many more by the mycobacterial proteins EsxQ, Apa, Rv1827, LpqN, Rv2074 and Rv1816 [160].

*Mycobacterium avium* subsp.* paratuberculosis*, also induces alternative splicing of 46.2% of all genes, including two genes responsible for monocyte to macrophage differentiation-associated maturation and lysosome function. Since their splice variants lead to failure of macrophage maturation, bacterial intracellular persistence is improved in the early phase of infection by hampered clearance [161]. Furthermore, alternative splicing of RAB8B, a key regulator of phagosome maturation, induced by *M. tuberculosis* infected cells leads to the production of a truncated protein. The alternative splicing event results in nonsense-mediated decay of RAB8B mRNA resulting in lowered protein levels, that dampens phagosome maturation [156].

Analysis of RNAseq data revealed that specific genes are chosen by pathogens for the manipulation of alternative splicing. Indeed, most dominant isoforms of protein kinases produced end up with the loss of critical functional domains including kinase domain or protein–protein interaction domains like SH2, SH3 and PH domains [162]. Considering nonsense-mediated decay of RAB8B mRNA after *M. tuberculosis* infection, it is concluded that this mechanism describes the two strategies host and bacteria developed during their evolutionary concurrence [156]. In this theory, increase of transcription after infection represents the host response, whereas splicing into a truncated isoform, which is destinated for decay, exemplifies bacterial interference. The exact mechanism how bacteria manipulate alternative splicing is not clear yet. Possibly, they activate cryptic or weak splice sites in the host genome to alter the splicing pattern but this has still to be proven [149].

However, another aspect that needs to be considered is the high diversity with that individuals react to the same pathogenic agent. For example, only 5–10% of the individuals in tuberculosis endemic countries that had contact with *M. tuberculosis* develop disease, whereas the majority either eliminates the pathogen or controls it in a metabolically altered latent phase [163,164]. An explanation are single-nucleotide polymorphisms (SNPs) that disrupt splice-site consensus sequences in 15% of human disorders induced by inherited point mutations, whose influence induce strongly fluctuating pathological conditions after varying activation of disease associated genes [165,166,167]. This was already reported for diseases as diabetes and seems to be also true for infections, as several SNPs were identified that change the host susceptibility to *M. tuberculosis* infections for example, in the intronic region of human ASAP1 (dendritic cell migration). Another polymorphism in IL-7RA helps to protect against tuberculosis due to an impaired IL-7Ra splicing [168,169]. Therefore, alternative splicing gets into focus of possible medical therapy developing splicing inhibitors that are already tested for cancer [149,170,171]. Nevertheless, the knowledge about alternative splicing during bacterial infections and their interplay is very limited and deserves more attention, as this could give more insights in individual susceptibility and immunity.

### 2.4. The Advantage of Modulating Host RNA Stability and Degradation

The lifetime of mRNAs has a major impact on the amount of proteins that can be produced; the shorter an mRNA is present, the less it can be transcribed. Since the lifetime of an mRNA is dependent on its stability, there are mechanisms to increase resistance to degrading RNases [11,12,13]. First, the 5′m^7^G cap and the 3′poly-A tail that are established post transcriptionally at all mRNAs but especially the length of the poly A tail can vary between mRNAs, determining the duration of resistance against enzymatic degradation [9]. Besides, these structures are involved in virus clearance, as viral mRNAs lack these structures, what marks them for RNA degradation machinery [172]. In addition, the structure of the mRNA alone influences its stability, as hairpin-structures and other secondary structures are formed dependent on the sequence and therefore, increase the stability [11,12,13]. In the following paragraphs we want to highlight the regulation of mRNA stability and decay mediated by miRNAs during infections and how bacteria interfere with this part of the host immune defense.

#### Manipulation of miRNAs to Favor Bacterial Survival

Their physiological properties enable non-coding RNAs (ncRNAs) to base-pair with their targets and interfere with a twofold effect on gene expression—one single ncRNA can bind multiple targets, thereby influencing several pathways at once and one gene can be regulated by several ncRNA fine tuning gene expression [173]. miRNAs are involved in several cellular processes as cell proliferation or differentiation and after studies with human monocytes treated with LPS, miR-146 was identified as anti-inflammatory miRNA proving miRNA involvement in inflammation [174,175,176]. Indeed, subsequent studies revealed specific expression of miRNA sets including miR-155, miR-146, let-7 and miR-29 (see Table 3) due to infection with different bacterial pathogens regulated in a time dependent manner [177,178,179,180,181]. Another study with dendritic cells infected with six different bacteria demonstrated a core infectious response in a temporal manner including 49 miRNAs that were always expressed and may, therefore, play essential roles in infectious responses. Additionally expressed miRNAs might hint towards specific variability and signatures arising from the individual pathogens [182]. Interestingly, following infection, the proportion of miRNA variants, the so called isomiRs, varies, which is supposed to effect miRNA identity and regulatory potential but has not been proven yet [182].

The induction of miRNAs is often dependent on PRR and NF-κB pathway induction in response to bacterial stimuli as LPS. Interestingly, there is a dose dependent reaction to the stimulus, as a low dose activates miR-146 that acts as anti-inflammatory regulator promoting tolerance to low doses by targeting two members of the NF-κB pathway, TRAF6 (TNF Receptor-associated factor 6) and IRAK1 (IL-1R-associated kinase 1) [176,183]. In contrast, at high doses of LPS, TNF-α and Interferon β induce miR-155 via TAB2 to maintain the proinflammatory NF-κB activity fighting pathogens and exerting a negative feedback on the immune system. Therefore, both mi-RNAs protect the host from sepsis and overreaction [183,184] but in a dose dependant manner. miR-155 is also involved in T helper cell development or promoting autophagy by inhibition of the mTOR (mammalian/mechanistic target of rapamycin) pathway, indicating that it represents an important part of an efficient immune response [185,186,187]. Actually, upon *Citrobacter rodentium* or *L. monocytogenes* infection, miR-155 null mice showed slower clearance and an impaired CD8^+^ T-cell response, respectively and miR-155 was identified as an essential factor during the vaccination process against *S.* Typhimurium [188,189,190,191].

Another miRNA family, the let-7 family, is repressed during infection or exposure to LPS, as Lin-28B expression is induced in a NF-κB dependent manner that blocks let-7 maturation [154,192,193,195,199,200]. Additionally, active repression of these miRNAs by bacteria has been reported in several studies (see Table 3). Many other bacteria induce miRNA expression and manipulate expression of these immune regulators in their favor (summarized in Table 4.)

In addition to *H. pylori* with miRNAs (see Table 4), there are two more interactions described, which depend on the effector CagA that activates NF-κB pathway. This effector induces an increased expression of miR-1289 that in turn leads to a decreased gastric acidity, as miR-1289 targets HKα, a component of the gastric H^+^/K^+^ ATPase [213]. Furthermore, Cag A induces cell cycle arrest at G1/S transition, which inhibits the renewal of the gastric epithelium, supporting *H. pylori* colonization [214]. In this context, miR372 and miR-373 expression is suppressed by Cag A, whereas miR-584 and miR-1290 expression is promoted. The latter target FOXA1, a negative regulator of the epithelial-mesenchymal transition, for inhibition, thereby favoring bacterial persistence and survival within the gastric epithelium [215]. Moreover, CagA suppresses miR320 and miR370 expression, who induce MCL1 (an anti-apoptotic gene) or downregulate FoxM1 expression, respectively. FoxM1 downregulation in turn activates p27^K1P1^ leading to cell cycle inhibition. Together, these factors decrease apoptosis and favor cell proliferation, which can lead to tumor development.

The *M. tuberculosis* effector ESAT-6 is also capable of manipulating miRNA expression to the benefit of the bacterium [212]. ESAT-6 downregulates let-7f expression in macrophages, leading to an enrichment of the deubiquitinating enzyme A20 that negatively regulates the NF-κB pathway. Furthermore, miR-155 expression is stimulated, resulting in BACH1 (a transcription regulator protein) and SHIP1 (SH-2 containing inositol 5′ polyphosphatase 1, a multifunctional protein expressed predominantly by hematopoietic cells) repression that in turn induces heme oxygenase 1 expression. Concurrently, serine/threonine kinase AKT is activated fostering bacterial dormancy and survival. This is subsidized by downregulation of SOCS1 (suppressor of cytokine signaling 1) and targeting of Rheb (Ras homolog enriched in brain), which is followed by macrophage apoptosis [199,210,211].

Evidence exists that bacteria are able to also produce their own regulatory RNAs that interfere with the host. In 28 bacterial genomes 68 possible candidate bacterial RNAs were found during an in silico search, which harbor secondary structures that could form miRNAs after host procession and be involved in 47 human diseases [216]. As an example, after *E. coli* ingestion *che-2* and *F42G9.6* gene expression was modulated and probably degraded in *Caenorhabditis elegans* by *E. coli* OxyS and DsrA ncRNAs [217]. Furthermore, *Mycobacterium marinum* expresses a pre-miRNA that associates with the host RISC complex in its mature, 23 nucleotide long form and could effectively downregulate its target mRNA when overexpressed [218].

It has been reported that exosomal transfer of host cell miRNAs is used to spread the host response to other cells and the ratios of miRNAs transported differ in a time and bacterial dependent manner. Hence, the ratios of miR-146a and miR-155 in exosomes can subsequently modulate host cell responses, favoring inflammation or recovery, respectively. The use of exosomes containing miRNAs could give rise for therapeutic possibilities to treat inflammation or to be used during vaccinations [219,220,221,222,223,224,225,226]. Moreover, exosomes containing miRNAs could be used for diagnosis, since they can be detected in many sample types (blood fluids, saliva, tears, urine, amniotic fluid, colostrum, breast milk, stool, etc.) and since there are unique patterns for each pathogen [219,220,221].

Regulation of gene expression through ncRNAs as miRNAs happens more immediately and flexibly than through transcriptional regulators [33]. The faster response is enabled by the cell and tissue type specific differentially regulated reservoir of ncRNAs, which also allows a precise fine-tuning of gene expression to organize immune defense and damage protection [33]. Taken together, investigation of the host-pathogen crosstalk with a focus on miRNAs and their usage and manipulation by bacteria provides new perspectives to fight bacterial mediated diseases.

### 2.5. Controlling Host Translation Improves Bacterial Persistance

Translation is a major regulator of gene expression and immune response. As many factors are needed to induce Ribosome association, start of translation and ongoing elongation, many ways exist to regulate or interfere with the translation machinery. Not surprising, inhibition of translation is a well-known strategy followed by bacteria to circumvent immune defense [227].

In most cases, eukaryotic translation is controlled during initiation, when ribosomes are recruited to the mRNA mediated by eukaryotic initiation factor 4F (eIF4F) that recognizes the 5′ cap structure with the help of its cap-binding subunit eIF4E [228]. The reversible association of this subunit with 4E-binding proteins (4E-BPs) inhibits the assembly of eIF4F and its release and activation are in turn mediated by the phosphorylation of the 4E-BPs [229,230,231]. This phosphorylation is induced by the serine/threonine kinase mTOR complex 1 (mTORC1), thereby requiring the protein Raptor for mTOR substrate binding whereas rapamycin binding inhibits phosphorylation and dissociation [232,233].

*Legionella pneumophila* (*L. pneumophila*) expresses five effectors, Lgt1, Lgt2, Lgt3, SidI and SidL, involved in global protein translation inhibition by interference with the eukaryotic elongation factors eELF1A and eELF1Bγ [227,234,235,236]. Moreover, *L. pneumophila* was also shown to negatively influence cap-dependent translation initiation mediated by ubiquitination of the mTOR pathway leading to suppression of the eukaryotic initiation factor 4E (eIF4E) through decreased eIF4E assembly into the translation initiation complex eIF4F [237].

Finally, the synthesis of IκB, an inhibitor of the NF-κB TF, is inhibited by *L. pneumophila.* This leads to a prolonged NF-κB activation resulting in the so-called effector-triggered response (ETR) including transcription of target genes, such as *Il23a* and *Csf2* that create a more pro-inflammatory state. Fascinatingly, mutants lacking effectors or the Dot/Icm type IV secretion system transferring them, still inhibit host translation via TLRs and NF-κB activation but not sufficient enough to fully induce ETR [234,235]. In addition, macrophages lacking TLR and Nod signaling still mediated MAPK signaling after exposure to the five *L. pneumophila* effectors that leads to inhibition of host translation. Therefore, translational inhibition does not exclusively rely on ETR but also on effector independent mechanisms that induce mTOR pathway downregulation and cytokine biasing [237]. In this context it is suggested that the host immune system senses not only for PAMPs but also for pathogen-encoded enzymatic activities that disrupt crucial cellular processes [227]. Interestingly, even if host translation is inhibited at the stage of initiation and elongation by *L. pneumophila*, there is still an inflammatory response detectable. The immune response is quite weak compared to its normal potential but few inflammatory cytokines, as IL-1α and IL-1β, circumvent translational inhibition by *L. pneumophila*, which is mediated by MyD88 signaling [227]. This demonstrates that the interference with cap-dependent host translation results in promotion of host defense, as highly abundant transcripts, which often encode proinflammatory cytokines, are favored for translation. Thereby, the bacterial benefit through blockage of host translation may consist of increased availability of amino acid nutrients beneath the dampened immune response, which is impeded by the host [234].

Translational inhibition is also known for *P. aeruginosa* infections in *C. elegans,* where Exotoxin A after its endocytosis into intestinal cells leads to suppression of elongation factor 2 (EF2), followed by selective translation of ZIP-2 and thus, induction of pathogen clearance [238,239]. As the 5′ UTR of zip-2 contains several untranslated ORFs (uORFs), it was proposed that the uORFs could mediate selective translation. Moreover, inhibition of translation by pharmacological inhibitors also caused induction of various stress response genes including *Il6*, *Il23*, *Il1α* and *Il1β* [240,241,242]. As these cytokines are still expressed when the elongation machinery is attacked by *P. aeruginosa* with Exotoxin A, a consideration of similar functionalities of the 5′ UTR or the 3′ UTR of cytokine genes to uORFs was raised. However, the ADP-ribosylation of elongation factor 2 (EF2) in host cells also triggered a strong immune response that is supposed to be the result of a conserved surveillance mechanism in response to inhibition of the translation elongation machinery [236,243]. Thus, a set of elongation factors can be considered, that are resistant to modification by these effectors or are at least not targeted. Therefore, future research is needed to get more information about the underlying mechanisms and potential therapeutic targets [227].

### 2.6. Modification of Protein Degradation/Activity as Last Possibility to Evade Host Immune Defense?

#### 2.6.1. Mechanisms to Interfere with Protein Degradation

To maintain cellular homeostasis, the quality of synthesized proteins must be controlled for a proper folding and products with quality issues must be degraded in a controlled manner. This quality control is taking place in the cytosol or the ER lumen, where chaperones and heat shock proteins catalyze and stabilize the protein folding [244]. In the case of physiological stress caused by DNA damage, chemical stimuli or pathogens, the ratio of misfolded or unfolded proteins in the ER lumen increases, causing additional stress. Then, the unfolded protein response (UPR), an evolutionary conserved signaling network, is activated resulting in downregulation of overall protein synthesis, except for chaperones and induction of ER associated protein degradation (ERAD [245]. The main kinases controlling the UPR, the inositol-requiring protein 1 (IRE1), PKR like ER kinase (PERK) and activating TF 6 (ATF6) are located inside the ER membranes. Their luminal domains bind the ER chaperone immunoglobulin binding protein (BiP) in unstressed conditions but in case of stress, BiP is released causing activation of the receptor proteins (IRE, PERKI and ATF6) [246]. Following BiP dissociation, oligomerization and autophosphorylation, the cytosolic RNase domain of IRE1 is activated targeting X-box-binding protein 1 mRNA (XBP1u) and transfers it into its spliced form (XPB1s). This enables transcription of the TF responsible for upregulation of UPR target genes fostering ERAD and enhances overall ER protein folding capacity [240,241,242]. The ER transmembrane kinase PERK1 also oligomerizes and autophosphorylates after activation, catalyzing the phosphorylation of the α-subunit of the eukaryotic initiation factor 2 (eIF2). This is followed by downregulation of global mRNA translation to reduce ER stress but favoring translation of some mRNAs as ATF4. ATF4 in turn induces several UPR target genes including C/EBP homologous protein (CHOP) [247,248]. In contrast to IRE1 and PERK, ATF6 translocates to the Golgi followed by its activation after proteolytical cleavage and activation of the b-ZIP TF to induce UPR target genes.

Since permanent ER-stress, which cannot be solved by UPR and ERAD, will finally induce apoptosis, intracellular bacteria have evolved strategies to interfere with those pathways [249,250,251]. Surprisingly, induction of UPR pathways can also promote bacterial replication, as bacterial effectors have been detected that induce UPR [19]. As result, it is difficult to refer an upregulation of UPR to bacteria using the increased ER folding capacity for their benefit or to the defense system of the host. In the case of *L. monocytogenes* infection, the effector LLO activates all three UPR pathways leading to induction of ER stress and reduction of bacterial survival. In contrast, the pharmacological block of UPR during infection reduced the intracellular replication of *Brucella melitensis* and *Brucella abortus* significantly [252].

*B. melitensis* and *B. abortus* both induce the IRE1 branch, a process often mediated by TLRs. The TLR adapter protein myeloid differentiation primary response gene 88 (MyD88) than mediates XBP1u splicing. Interestingly, the bacterial effector protein TcpB, secreted by *B. melitensis*, is able to induce the UPR target genes BiP, CHOP and ER DnaJ-like 4 (ERdj4) in a MyD88 independent manner; instead, the TcpB protein itself is required for UPR induction [253].

The induction of IRE1 by *B. abortus* is mediated by the secreted effector VceC after binding of BiP inside the ER lumen. This is followed by IRE1 dependent activation of Nod1/Nod2 innate immune signaling resulting in proinflammatory cytokine expression [254,255]. The ectopic expression of VceC leads to the structural reorganization of the ER and IL-6 production is stalled after infection with *B. abortus* vceC mutants unable to induce UPR. In mice infected with *B. abortus* vceC mutants necrosis was reduced and survival of the pups was increased [255]. Following infection with *B. abortus* WT, similar effects were observed after treatment with the general UPR inhibitor tauroursodeoxycholic acid, leading to the assumption that pharmacological UPR inhibition could be a promising treatment of *B. abortus* infections.

Several other bacteria are known to inhibit the UPR pathway, as it represents a major host defense mechanism involved in bacterial sensing mediated by TLR signaling. Some examples are summed up in Table 5 but unfortunately, the underlying mechanisms are rarely understood. In some cases, as for *L. pneumophila,* it is known that the observed downregulation of UPR is effector dependent, as mutants lacking functional T4SS were unable to induce those changes but the effectors and its targets have not been identified yet [256,257].

Taken together, bacteria follow different strategies of interference with protein folding to ensure their intracellular survival. Manipulation of UPR is one strategy to achieve the best outcome for bacteria by either activation or inhibition of UPR. Activation may be induced to take advantage from increased protein folding capacity and lipid biosynthesis by host cells, whereas UPR blockage should hamper host defense, such as apoptosis or innate immunity [261]. In this context, further investigations are needed to understand the underlying mechanisms, how bacterial pathogens manipulate the UPR and which strategy is favored by the individual pathogens in a spatially and temporally manner.

#### 2.6.2. Control of Protein Activity enables Bacteria to Direct Host Immune Reaction

As already mentioned above, the final level to regulate gene expression, for example, to avoid prolonged inflammatory response, is to control the activity of proteins followed by their degradation. The mechanisms to influence protein activity or to mark a protein for degradation are mediated via attachment of functional groups. To ensure a precise signal transduction, proteins are activated in most cases by addition of phosphate groups, which must be removed when the inducing stimulus is ending. The dephosphorylation of MAPKs and the resulting interruption of host signaling cascades, leading to reduced inflammation and an increase of bacterial replication inside the host, is a common bacterial strategy, for example, used by *S.* Typhimurium by secretion of the effector SpvC, a phosphothreonine lyase [262].

In addition, protein activity, subcellular localization and stability is not only regulated by ubiquitination or phosphorylation but also by reversible acetylation which is proposed for approximately 1700 proteins [263]. These include TFs, structural proteins and signal transduction regulators, indicating that reversible acetylation is critical for cell homeostasis [264]. As there are many examples for this kind of activation of proteins (e.g., histones) mentioned in the chapters above, the focus will here lie on the description of the signaling mechanism by ubiquitination. An ATP-dependent enzymatic cascade establishes covalent ubiquitin attachment to proteins, mediated by enzymes that activate ubiquitin (E1), conjugate ubiquitin (E2) and ligate ubiquitin (E3). Ubiquitin can be ubiquitinated at seven distinct lysine residues and is able to form linear or branched chains; the degree and the linkage determine, whether the substrate is degraded or associated with cell signaling [265]. The established ubiquitin modifications can be removed and modified by deubiquitinases (DUBs) to change the linkage pattern and, as consequence, the destination of the substrate.

Intracellular bacteria are able to mimic enzymes involved in ubiquitination processes, especially DUBs and E3 ubiquitin ligases, to modulate the ubiquitin pathway [266,267]. The SidE effector family (SidE, SdeA, SdeB and SdeC), secreted by intracellular bacteria as *L. pneumophila*, possesses domains conferring multiple enzymatic functions used for ubiquitination into a single effector without the requirement of ATP. Thereby, the mono-ADP-ribosyltransferase domain of SidE family members modifies host ubiquitin posttranslationally by attachment of phosphoribose on arginine 42 to generate ADP-ribosylated ubiquitin (ADPR-Ub) [268,269,270,271]. This intermediate is than cleaved by nucleotidase/phosphohydrolase/phosphodiesterase domains into AMP and phosphoribosylated ubiquitin (PR-Ub), which in the next step is attached to host proteins via a noncanonical serine-linked phosphodiester bond. The DUB domain (also found in SidE family members) than removes host ubiquitin modifications but not the SidE-mediated ubiquitination and reduces the level of ubiquitination on the surface of the *L. pneumophila*-containing vacuole (LCV) [271]. In addition, overexpression of SidE effector family members in mammalian or yeast cells generates a pool of PR-Ub that interfere with E1 and E2 enzymes hampering conventional host ubiquitination pathways. This results in interruption of mitophagy, immunity, as shown for TNF-induced NF-κB nuclear translocation, proteasomal degradation (for example of hypoxia inducing factor 1α) and other ubiquitin-regulated processes in the host [268,269]. Furthermore, ER structure and host membrane trafficking are modulated by SidE effector family members to enable LCV biogenesis. To ensure precise temporal control over the signal transduction mediated by ubiquitination, *L. pneumophila* secretes another DUB effector, SidJ, that is able to remove ubiquitin modifications established by the SidE effector family and the mammalian ubiquitination machinery [268,269,272].

*S. flexneri* also owns an effector with E3 ubiquitin ligase activity, IpaH9.8, that disrupts the NF-κB dependent pathway in the cytosol and impairs the activity of U2AF35, an mRNA splicing factor, leading to host inflammatory responses being suppressed [273,274,275]. The effector owns an N-terminal domain containing Leucine-Rich Repeats (LRR), also known as the LPX-domain, that is responsible for substrate recognition and a C-terminal E3 ubiquitin ligase domain, referred to as NEL domain (novel E3 ligase). Notably, the original structure of this domain differs from known eukaryotic E3 ligases, therefore, IpaH9.8 and its orthologues in other bacteria, for example, SspH 1 of *Salmonella enterica*, are part of a novel family of bacterial E3 ubiquitin ligases [276,277,278,279,280,281]. SspH 1 targets the host kinase PKN1 for proteasomal degradation, thereby functioning as ubiquitin ligase. In this context, it inhibits NF-κB dependent pro-inflammatory genes and regulates activation and function of neutrophils and macrophages as part of the androgen receptor pathway [278].

Enterohaemorrhagic *E. coli* express the T3SS effector protein NleG5-1, which contains a ubiquitin ligase U-box domain for ubiquitination and degradation of nuclear proteins. One target is a member of the mediator complex, MED15 that is a master regulator of RNA Pol II- dependent transcription [282]. The Ank-family expressed by *Orienta tsutsugamushi* (causative reagent of scrub typhus) is another protein family involved in ubiquitination and degradation, characterized by N-terminal Ank repeats and a C-terminal F-box-like domain termed as PRANC (pox protein repeats of ankyrin C terminus) motif. This family includes the proteins 1U5, 1A, 1B, 1E, 1F, 1U4 and 1U9 that interact with two members of multiprotein E3 ubiquitin ligase complexes, CULLIN-1 and SKP1 [283,284]. These proteins are supposed to act as mediators, as the ANK domain shall specifically bind target substrates, while the F-box recruits SKP1 promoting complex formation and finally inducing substrate degradation. This is also suggested for the degradation of the TF EF1α, probably induced via function of Ank 1U5 [283].

Especially the ubiquitin–proteasome system is a favored target of bacterial pathogens to manipulate the host cell cycle (summarized in Table 6). Two prominent targets are Skp1–Cullin1–F-Box protein (SCF, active throughout the cell cycle) and Anaphase-Promoting Complex/Cyclosome (APC/C, only regulatory active during Mitosis and late G1 phase), two classes of E3 ubiquitin ligase complexes inducing the degradation of cell cycle key regulators, thereby promoting its progression. A RING finger protein within both complexes enables binding to a scaffold cullin-like protein, the ubiquitin conjugating enzyme (E2) and distinct substrate-binding subunits. SCF are rated among the large family of Cullin-Ring E3 ubiquitin ligases (CRLs) as they are regulated via conjugation/deconjugation of the ubiquitin-like protein NEDD8 at the cullin subunit of SCF [285,286,287,288,289].

Diverse animal pathogens, such as *E. coli*, *Yersinia pseudotuberculosis*, *Burkolderia pseudomallei* and *Photorhabdus* spp., express so-called Cycle inhibiting factors (Cif) that target SCFs and CRLs resulting in cell cycle arrest [292]. This effector expressed by *E. coli*, Cif_Ec_, induces cell cycle arrest at G1/S and G2/M transitions by accumulation of cyclin dependent kinase inhibitors p27/_Kip1_ and p21_Waf1/Cip1_ and inhibition of key kinases. The appearing cytopathic effect is accompanied by cell enlargement and production of actin stress fibers.

Delay of mitotic progression was observed when IpaB, an effector of *S. flexneri*, induced unscheduled APC/C_cdh1_ activation and degradation of its substrates [293]. IpaB has to localize to the nucleus during G2/M phase and has to bind Mad2L2/MAD2B, the mitotic spindle assembly checkpoint protein that inhibits APC/C_cdh1_. The benefit of these mechanisms for the bacteria is not completely clarified yet but it is suggested that delay of cell renewal and cell turnover enables the bacteria to persist and further colonize the host tissues.

## 3. Conclusions

During the long coevolution of pathogenic bacteria and their host cells, several strategies developed on both sides to fight their counterpart and keep predominance. Thus, intracellular bacteria reach for the establishment of an intracellular niche that allows survival, replication and persistence. This state is achieved through disarming of the host immune defense while keeping it healthy enough to gain permanent nutrition excess. Bacterial effectors are crucial tools during the whole process, thereby targeting the host immune response at each level of gene expression. Even, if there are already many bacterial effectors reported mimicking host enzymes or featuring novel enzymatic functions, the complex interaction mechanisms and networks are still not completely understood. Moreover, it appears, that pathogenic bacteria can target different pathways simultaneously or one pathway with several effectors, thereby creating a species-specific modification pattern. Therefore, understanding the individual strategies gain new insights into the complex host-pathogen interactions during infections. Further investigation might unravel, which bacterial strategies are more efficient or where host cells already found strategies to circumvent attempts of bacterial manipulation. This highlights the importance of further research on bacterial subversion of the host immune response considering each level of gene expression, as new promising targets for successful bacterial clearance during medical therapy might emerge.

## Figures and Tables

**Figure 1 ijms-21-03730-f001:**
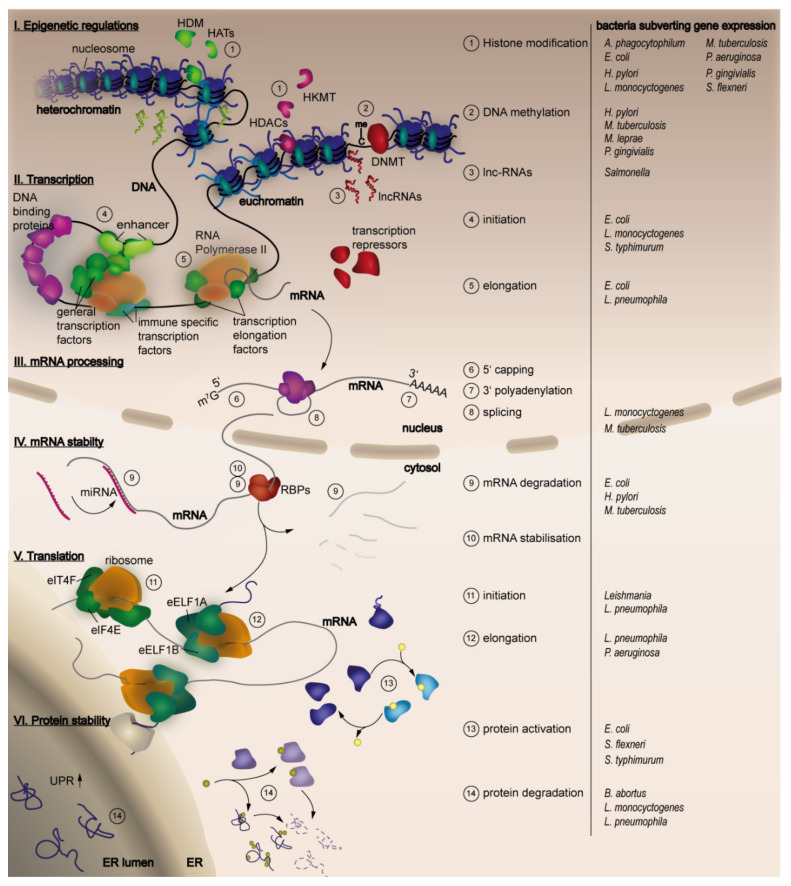
Steps of host gene expression manipulated by bacterial pathogens. The figure provides an overview over the main steps of gene expression that are indicated at the left side (I-VI). The numbers in the scheme highlight distinct characteristic processes that are part of each gene expression level and are listed in the legend at the right side. Different bacterial pathogens (indicated at the right) have been described to target the distinct steps and processes during host gene expression to their favor. For detailed information please refer to the text of this review.

**Table 1 ijms-21-03730-t001:** Histone modifications established by bacterial effectors.

Target	Modification	Bacterium	Effector	Mediator	Cellular Function	References
H3S10	de- phosphorylation	*L. monocytogenes*	LLO	unknown	Reduced expression of important immune regulators	[36]
H3S10	de- phosphorylation	*S. flexneri*	OspF	MAPK		[39,49,50]
H3S10	de- phosphorylation	*Clostridium perfringens*	perifringo- lysin	unknown	unknown	[36]
H3S10	de- phosphorylation	*Streptococcus pneumoniae*	pneumolysin	unknown	unknown	[36]
H3K18	deacetylation	*L. monocytogenes*	InlB	c-Met induced SIRT2 recruitment PI3K/AKT	Reduced expression of important immune regulators	[36]
H3K4	methylation	*L. pneumophila Philadelphia LP02*	LegAS4	direct	Transcriptional activation of ribosomal genes	[49]
H3K4	methylation	*B. thaliandensis*	BtSET	direct/NF-κΒ	Transcription of rRNA genes	[45,47]
H3K9	acetylation	*M. tuberculosis*	Rv3423.1	maybe direct		[51]
H3K14	methylation	*L. pneumophila Paris*	RomA	direct	Transcriptional repression	[46]
	acteylation	*M. tuberculosis*	Rv3423.1	maybe direct		[51]
H3K23	Global deacetylation	*H. pylori*		H3S10 dephosphorylation	Differential c-Jun and HSP70 expression	[52]
H1	trimethylation	*B. anthracis*	BaSET	direct/NF-κΒ	Transcriptional repression of NF-κΒ target genes	[47]
H2B, H3, H4	methylation	*C. trachomatis*	NUE	direct	Transcriptional repression	[53]
H3, H4	acetylation	*P. gingivivalis*	SCAFs, LPS	unknown	unknown	
H3	deacetylation	*M. tuberculosis*	unknown	HDAC1	Silencing of inteleukin-12β, suppression of T-helper1 response	[54]
H4	deacetylation	*L. monocytogenes*	LLO	H3S10 de- phosphory- lation	unknown	[36]
H4	deacetylation	*M. tuberculosis*	unknown	HDAC complex containing mammalian co-repressor Sin3A	Inhibition of interferon-γ- dependent HLA-DR gene expression	[55]

**Table 2 ijms-21-03730-t002:** Bacteria targeting histone modifying enzymes.

Target	Modification	Bacterium	Effector	Cellular function	References
**HDAC1**	induction	*A. phagocytophilum*	Ankyrin A	suppression of target genes as CYBB that encodes cytochrome b-245, beta polypeptide	[56]
**HDAC1**	induction	*P. aeruginosa*	2-Amnoacetophenone	reduced expression of inflammatory cytokines and chemokines	[65]
**HDAC1**	induction	*M. tuberculosis*	unknown	Silencing of inteleukin-12β, suppression of T helper1 response	[55]
**HDAC2**	repression	*P. gingivalis*	SCAFs	Activation of genes	[68,72,73,74,75]
**P300**	repression	*E. coli*	Proteinase NleC	decreased IL-8 production	[76]
**HKMT**	repression	*P. gingivalis*	SCAFs	Inhibition of heterochromatin marks	[73,74,75]

**Table 3 ijms-21-03730-t003:** Activity of miRNAs in the host response. The arrows indicate changes of miRNA expression induced after bacterial infection that result in the described alterations..

miRNA	Target	Cellular Function/Cells Involved	Induced Changes During Infection	Described for Infection with	References
let-7B	TLR4	TLR sensing	↓ Promoted TLR sensing and NF-κΒ activity	several bacteria	[192]
let-7A, let-7D	IL-6, IL-10	Pro-inflammatory and anti-inflammatory cytokine	↓ Maintaining balanced immune response	several bacteria	[193]
let-7C	mTOR-pathway		↓ Modulation of T-cell activity	several bacteria	[194]
let-7I	SNAP23	Exosome release	↓ Antimicrobial response, cell to cell communication	several bacteria	[195]
miR-29	IFN-γ	Immune response to intracellular bacteria/NK-cells, CD4^+^ and CD8^+^ T-cells	↓ increased IFN-γ expression and bacterial clearance	*L. monocytogenes, Mycobacterium bovis* bacillus Calmette-Guérin (BCG)	[196]
miR-192, miR-378, miR-215, miR-148A, miR-200C, miR-200B	zinc finger E-box–binding homeobox ZEB1 and ZEB2 (transcriptional repressors of E-cadherin)	Intestinal homeostasis, gut transcriptome, tissue integrity, immunity and metabolism	↓	*L. monocytogenes*	[197,198]

**Table 4 ijms-21-03730-t004:** Bacteria manipulating host miRNA to circumvent immune response. Arrows indicate an up or downregulation and refer to the miRNA before.

Bacterium	miRNA	Cellular Function	miRNA Target	miRNA Expression Induced by	Bacterial Benefit	References
*L. monocytogenes*	miR-29 ↑	Expression of IFN-γ by NK cells	IFN-γ mRNA		unknown	[196]
*Salmonella*	miR-30c, miR-30e ↑	Host SUMOylation	Ubc9 (cellular E2 SUMO- conjugating enzyme)		Reduction of host SUMOylation	[201]
	miR-15 family ↓	Cell cycle	Cyclin D1	Inhibition of tran- scriptional factor E2F1 production	De-repression of cyclin D1,G1/S cell cycle transition, bacterial intracellular replication	[202]
	miR-128 ↑	Recruitment/ activation of macrophages	Macrophage colony- stimulating factor (M-CSF)	p53 signalling pathway	Impairment of M-CSF mediated macrophage recruitment	[203]
*H. pylori*	miR-21, miR-222 ↑		tumour suppressor RECK			[204,205]
	miR-30b ↑	Formation and maturation of autophago- somes	BECN1 and ATG12 transcripts		Autophagy interference, intracellular persistence and survival	[206]
*M. tuberculosis*	miR-132, miR-26a	Recruitment/ activation of macrophages	Tran- scriptional coactivator p 300		Downregulation of INF-γ signalling cascade, limitation of macrophage response to INF-γ signalling	[207]
	miR-125b, miR-155, miR-99b	Recruitment/ activation of macrophages	TNF-α		Reduction of TNF-α production, increase of SHIP1 transcription, reduced macrophage recruitment	[208,209]
	miR-155↑	Recruitment/ activation of macrophages and macrophage apoptosis	Repressing BACH1, SHIP1 and SOCS1, Rheb		Induction of heme oxygenase 1 expression, activation of serine/threonine kinase AKT	[199,210,211]
	let-7f↓	Inhibitor of NF-κΒ pathway	A20 (deubiquitinating enzyme)		favored bacterial survival in macrophages	[212]

**Table 5 ijms-21-03730-t005:** Inhibition of unfolded protein response (UPR) by bacteria.

Effector	Bacterium	Target	Cellular Function	Manipulation	Reference
glucosyltransferase effector proteins	*L. pneumophila*	unknown	IRE1 branch of UPR	Inhibiting splicing of XBP1u mRNA	[256,257]
unknown	*L. pneumophila*	unknown	UPR	inhibit the translation of BiP and CHOP	[256,257,258]
unknown	*Simkania negevensis*	unknown	UPR	BiP induction (early), later on inhibition	[259]
unknown	*Simkania negevensis*	unknown	UPR	blocks the translocation of preexisting CHOP protein to the nucleus	[259]
unknown	*Simkania negevensis*	unknown	PERK branch of UPR	Reduced phosphorylated eIF2 levels	[259]
unknown	*L. monocytogenes*	unknown	PERK branch of UPR	Reduced phosphorylated eIF2 levels	[260]
unknown	*C. trachomatis*	unknown	PERK branch of UPR	Reduced phosphorylated eIF2 levels	[260]

**Table 6 ijms-21-03730-t006:** Bacteria manipulating the host cell cycle.

Effector	Bacterium	Target	Cellular Function	Manipulation	Reference
VIRF	*Agrobacterium*	SCF	cell-cycle progression by targeting key regulators for rapid degradation	Promoting cell proliferation	[290,291]
cycle inhibiting factors (CIF)	*E. coli* *(Cif_Ec_)* *Yersinia pseudotuberculosis* *(Cif_Yp_)* *Burkolderia pseudomallei (Cif_Bp_)*	SCF CRL	arrest of the cell cycle at G1 and G2 phases	Inhibition of host cell proliferation	[292]
IPAB	*S. flexneri*	Mad2L2/MAD2B	inhibitor of the APC/Ccdh1	cell cycle arrest	[293]

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
