# Peer review of "From Gene to Protein—How Bacterial Virulence Factors Manipulate Host Gene Expression During Infection"

_ijms, 2020, doi:10.3390/ijms21103730_

Round 1

Reviewer 1 Report

The manuscript (ID: IJMS-805204) presents a detailed review on the interactions between bacterial pathogens and a host with great emphasis on the influence of various bacterial effectors on different levels of host gene expression, including epigenetic regulations. The manuscript is interesting, is well written and should have a broad scientific impact. Although some minor corrections should be made.

Comments:

1) Figure 1 – It would be more legible if the numbering of reactions/processes in all steps (I-VI) is unified (from 1 to 14). Moreover, I can not find the number 3 (lnc-RNAs) in the step I of this scheme.

2) First sentences of all table and figure legends should start with a capital letter.

3) Line 89 – Nucleomodulins are a kind of bacterial effectors, so it would be better to write “…by nucleomodulins and other bacterial effectors.”

4) There is a lack of references to Tables 1, 2, 5 and 6 in the text.

5) In Tables 1, 2 and 4 – A caption of the column 3 should be “Bacterium”.

6) Line 264 – should be “…following uropathogenic E. coli …”

7) Line 272 – should be “Mycoplasma hyorhinis” (in italics).

8) Line 275 – should be “Mycobacterium tuberculosis” not “Mycoplasma tuberculosis”.

9) Lines 277-284 – It seems that the reference nr 88 is not appropriate for this part of the text.

10) Line 300 – reference?

11) Line 328 – should be “…40 canonical lncRNAs…”

12) Line 355 – should be “nucleomodulin OrfX”.

13) Line 367 and in other parts of the text – Currently, if Salmonella name refer to a name of serotype it should be written with a first letter capitalized and not italicized (Salmonella Typhimurium or Salmonella ser. Typhimurium). However, other designation, like Salmonella typhimurium, is also used in the literature.

14) Line 446 – “subsp.” should not be in italics.

15) Line 466 – should be “…in tuberculosis endemic countries…”.

16) Lines 591-604 – This part of the text refers to Leishmania spp., a parasite (not a bacterial pathogen), and does not fit in the manuscript matter, thus I propose to exclude it from the manuscript.

17) Line 701 – numbers of the references should be used.

18) Line 756 – “S. flexneri” should be in italics.

19) Line 763 – “Salmonella enterica” should be in italics (a species name).

20) Line 771 – “scrub typhus” would be better.

21) Page 25, a list of pathogens – should be “B. abortus - Brucella abortus” (not Bacillus)

Reviewer 2 Report

The paper entitled „Bacterial virulence factors manipulating host gene expression” by the authors is a comprehensive and current paper on the topic, which is definetly of interest to a readership of microbiologist, immunologist and other scientist. In addition, it will definetly garner additional citations to the journal.

Here are my comments:

General: bacterial names should not be designated as abbreviations in the text (in brackets after their full name and at the end of the text). The authors should adhere to the international conventions of writing bacterial names, i.e. write them in full on first mention and later on, only use the abbreviated form for the genus name and only add the species designation.

The text of the words could be more crisp and concise in some parts, therefore I suggest the authors to carefully read through the text and try to compact some sections as they are a bit wordy.

Title: I think the authors should think about a new title or complementing the current title, as it is very bland and not informative, and does not represent the text in many ways.

L28: of the adequate matching immune response

L37: path-specific adapter proteins

L44: on the chromatin level

The quality of Figure 1. in my opionion is inadequate: the image needs additional sharpening, in its current stage is not legible, the images and fonts are too small.

L106: posttranslational modifications

L148: produce toxins

L202-203: maybe a reference is missing? (german text)

L207-208: maybe a reference is missing? (german text)

L227: Consider including the following reference:

Antibiotics 2020, 9(4), 153; https://doi.org/10.3390/antibiotics9040153

L300-301: maybe a reference is missing? (german text)

L387: Consider including the following reference:

Medicina 2019, 55(7), 356; https://doi.org/10.3390/medicina55070356

Section 2.2.: P53 should be written p53

Table 4. Bacterial names should not be in capital letters here

L697: maybe a reference is missing? (german text)

„2.6.2. Control of protein activity, a smart way to direct host immune reaction” please rename this section to something more concise.

L756-767: bacterial names should be in italics.

L781-782: maybe a reference is missing? (german text)
